# Pneumatic Microballoons for Active Control of the Vibration-Induced Flow

**DOI:** 10.3390/mi14112010

**Published:** 2023-10-29

**Authors:** Taku Sato, Kanji Kaneko, Takeshi Hayakawa, Hiroaki Suzuki

**Affiliations:** Department of Precision Mechanics, Faculty of Science and Engineering, Chuo University, Tokyo 112-8551, Japan; t_sato@nano.mech.chuo-u.ac.jp (T.S.); kaneko@nano.mech.chuo-u.ac.jp (K.K.); hayaka-t@mech.chuo-u.ac.jp (T.H.)

**Keywords:** vibration-induced flow, active flow control, micropillars, microballoons, pneumatic actuator

## Abstract

Vibration-induced flow (VIF), in which a mean flow is induced around a microstructure by applying periodic vibrations, is increasingly used as an active flow-control technique at the microscale. In this study, we have developed a microdevice that actively controls the VIF patterns using elastic membrane protrusions (microballoons) actuated by pneumatic pressure. This device enables on-demand spatial and temporal fluid manipulation using a single device that cannot be achieved using a conventional fixed-structure arrangement. We successfully demonstrated that the device achieved displacements of up to 38 µm using the device within a pressure range of 0 to 30 kPa, indicating the suitability of the device for microfluidic applications. Using this active microballoon array, we demonstrated that the device can actively manipulate the flow field and induce swirling flows. Furthermore, we achieved selective actuation of the microballoon using this system. By applying air pressure from a multi-input channel system through a connection tube, the microballoons corresponding to each air channel can be selectively actuated. This enabled precise control of the flow field and periodic switching of the flow patterns using a single chip. In summary, the proposed microdevice provides active control of VIF patterns and has potential applications in advanced microfluidics, such as fluid mixing and particle manipulation.

## 1. Introduction

Microfluidics is a type of technology for manipulating fluids at the microscale, and various operations have been conducted depending on their applications [1,2]. Active flow control for cell/particle manipulation and fluid mixing within microenvironments often relies on external forces, such as electric [3], magnetic [4], and optical [5] techniques. However, although powerful, these systems are often burdened by the complexity of device fabrication and limited applicability, owing to the necessity of specific materials and mechanisms.

An alternative approach that has gained recognition in recent years is vibration-induced flow (VIF). VIF is a microscale fluid manipulation technique that is widely known for its simplicity [6,7,8,9,10]. It leverages the solid–liquid interaction induced by the vibration of microstructures on a substrate, resulting in localized flows around the structures from zero-mean vibration. Active control of the flow patterns can be achieved by modulating the vibration conditions, such as the amplitude, frequency, and shape/arrangement of the structure.

Thus far, various microstructural arrangements and vibration conditions have been investigated to achieve specific flow patterns for intended purposes. Simple changes in vibration significantly alter the flow pattern; for example, rectilinear vibration induces multiple vortices around the pillar, whereas circular vibration generates a swirling flow [11,12,13,14,15]. For applications such as cell trapping and manipulation, micropillars are distributed in a spiral arrangement to gather cells at the center of the spiral [6,7]. Further efforts have been made to expand various types of fluid control, such as periodic switching of flow patterns around asymmetric-shaped (ninja-star-shaped) pillars, enabling cell trapping and alignment on a single chip [16]. In addition, acoustically induced localized flow generated around sharp edges has been applied for efficient fluid mixing [17,18,19]. Instead of using solid microstructures, micro-air bubbles in water have also been used to induce microlocal flow through vibration [20]. This approach has been used in various applications, including mixing [21,22,23], the manipulation of microobjects [24], and size-dependent sorting [25,26]. Furthermore, efforts have been made to expand active fluid control by transporting bubbles using the electrowetting-on-dielectric technique [27].

These studies show that the flow pattern can be significantly altered by varying the vibration conditions and microstructure shape and arrangement. However, because the structure is fixed to the substrate, fluid or particle manipulation within a single device is limited. Interest in spatial and temporal structural deformations within relatively straightforward systems is growing to improve flow control versatility.

An established method for achieving spatial and temporal structural deformation is pneumatic actuation [28,29,30]. This approach involves altering the shape of the elastomer membranes by applying air pressure. Previous studies successfully integrated pneumatic actuators into microenvironments to control fluid flow, particularly through valve structures [31,32,33,34].

In this study, we developed a microdevice that actively controls VIF patterns using pneumatic actuation of protrusions (microballoons) as a structure to induce local flow. This advancement enables active flow control within a single device, spatially and temporally. We have successfully fabricated a microballoon device driven by air pressure and conducted a quantitative analysis of the resulting flow field. Furthermore, we performed spatial control of the flow resembling a blinking vortex flow; this serves as a model system for chaotic mixing by replicating the periodic switching of vortex positions [35,36].

## 2. Materials and Methods

### 2.1. Structure and Driving Principles of the Microballoon Device

Figure 1 shows a conceptual diagram of the device used for the selective actuation of a microballoon. This device consisted of three layers. The top layer was an Si wafer with through-holes over which the elastomer membrane was coated. The middle layer was a thin polydimethylsiloxane (PDMS) plate with a channel for delivering compressed air to the balloons. Finally, the bottom layer was a PDMS slab that served as the base structure to enable connections to the pressure tubes (Figure 1a). Figure 1b,c show the perspective and side views of the pneumatic actuation principle, respectively. Air pressure was applied to the inlet of the device through a connection tube. The applied pressure was transmitted to the top layer, causing the elastomer membrane to expand and form microballoons. Figure 1d shows the top view of the connection scheme between the microballoon device and the air-pressure pump. On the top layer (gray square in Figure 1d), microballoons were arranged in a 5 × 5 matrix, and each column of these microballoons was connected to an air channel in the middle layer. Five air channels were alternately connected to two air pressure inlets (inlets A and B) to achieve the selected actuation of the microballoon column. Thus, this design allows the actuation of the corresponding columns of the balloon structures, depending on the input channel of the air pressure.

### 2.2. Fabrication Processes for Consisting Layers

First, we fabricated a microballoon layer, as shown in Figure 2a. We used a one-sided mirror Si wafer with a thickness of 200 µm as a substrate. Silicone rubber resin (KE-1950-30, Shin-Etsu, Tokyo, Japan) was diluted with hexane at a volume ratio of 1:1.5 and spin-coated over the non-mirror side of the Si wafer at 2000 rpm for 30 s. Subsequently, it was cured by baking at 150 °C for 15 min on a hot plate. The thickness of the elastomer membrane (*t*) obtained was 13 µm. Next, AZ P4620 positive photoresist (AZ Electronic Materials) was spin-coated at 3000 rpm for 30 s on the mirror side of the wafer (the opposite side of the elastomer membrane), and a 5 × 5 array of circles with a 200 µm diameter and 300 µm spacing was exposed and developed using standard lithography. The selection of this specific diameter and spacing was made to enhance the efficacy of active control of microballoons and to ease the alignment of each layer in fabrication. Deep etching was then performed to form through-holes using a deep reactive-ion etching (DRIE) apparatus (RIE-400iPB, Samco, Kyoto, Japan) from the mirror side of the wafer. During this process, the Si wafer was transformed into a square with a side length of 10 mm (Appendix A). Finally, the photoresist residues were removed with acetone. An actual image of the elastomer-membrane-coated Si wafer with through-holes is shown on the right-hand side of Figure 2a.

Second, the middle layer with five air channels (50 µm width and 25 µm height) was fabricated (Figure 2b). For selective actuation, three channels at the center and both ends were connected to inlet A, whereas the other two channels were connected to inlet B (Figure 1d). A master mold of the air channel structures was fabricated with a negative photoresist (mr-DWL-40, Micro Resist Technology GmbH, Berlin, Germany) using direct lithography (DL-1000, Nanosystem Solutions, Okinawa, Japan). Next, 1.8 g of PDMS resin (SILPOT 184 W/C, Dow Corning Inc., Midland, MI, USA) was mixed with its curing agent in a 10:1 weight ratio and poured over the master mold. After curing at 85 °C for 120 min, the PDMS sheet was peeled off the mold. The PDMS sheet was cut to dimensions of 10 mm × 10 mm to correspond to the size of the diced Si wafer. Two vertical holes with diameters of 2 mm were created to allow the air pressure to reach the top layer.

Third, a bottom-base structure of the tube connection was fabricated (Figure 2c). This base structure was a 14 mm × 14 mm square with spacers of 1 mm in width and height at the four corners. Two squares, each with 2 mm sides on the surface, were aligned with the holes of the air channel sheet when stacked with them. The master mold of the base structure was fabricated using stereolithography (SPACE ART, Kantatsu, Tokyo, Japan). The surface of the master mold was coated with 0.75 µm thick Parylene C as a passivation layer. The PDMS resin was mixed with its curing agent at a weight ratio of 10:1 and poured over the master mold. After curing at 80 °C for 120 min, the PDMS slab was peeled off from the mold and cut with a knife. Finally, two lateral holes of 0.7 mm diameter were formed in the bottom base structure for tube connection using a biopsy punch.

### 2.3. Device Assembly

Three layers, fabricated as described in the previous section, were assembled. First, the Si wafer with an elastomer membrane and the air channel sheet were bonded. After applying oxygen plasma for 5 s on the mirror side of the Si wafer and the channel side of the PDMS sheet using a compact etcher (O_2_ flow rate: 20 mL/min, 75 W) (FA-1, Samco Inc., Kyoto, Japan), 5 μL of ethanol was dropped onto the bonding surface of the sheet. The Si wafer was then placed on a PDMS sheet before the ethanol dried. Because the sizes of both plates were identical, they aligned spontaneously via capillary action (Appendix A). This assembly was baked on a hot plate at 120 °C for 1 min to evaporate the ethanol so that it was covalently bonded. Subsequently, a weight of approximately 160 g was placed on top of the Si wafer side, and the bonding was completed by further baking at 120 °C for 2 h. Next, oxygen plasma was applied for 5 s with the same apparatus and conditions on both the bottom layer surface and the assembly of the air channel sheet/Si wafer. After manual alignment, the samples were pressed together. Bonding was completed by baking at approximately 120 °C on a hot plate for approximately 30 min. Through this bonding process, all the components were tightly assembled.

Finally, the device was connected to an air-pressure pump. One end of a 0.7 mm diameter tube was inserted into the holes in the bottom layer. The other ends of the tubes were connected via a connector to the three-channel pump with air pressure control (Onchip Droplet Generator, Onchip Biotechnologies Co. Ltd., Tokyo, Japan).

### 2.4. Actuation Test and Height Measurement of Balloon Actuators

The height of balloon actuation was measured. The balloons were observed from the side using an apparatus designed to measure the droplet contact angle (DropMaster DMs-401, Kyowa Interface Science Co., Ltd., Saitama, Japan), which could observe the balloon from a lateral angle (the experimental setup is shown in Appendix A). The device and connection tube were fixed to the stage. The device was set up in a slightly slanted position relative to the lateral objective lens by inserting a piece of double-sided tape on the back of the device to make the microballoons visible in the depth direction. The applied pressure was increased in 10 kPa increments from 0 to 30 kPa, and balloon expansion was observed. Additionally, the displacement of the balloons upon application of pressure was quantitatively measured using a laser scanning microscope (VK-9710, KEYENCE Inc., Osaka, Japan) (Appendix A). In the quantitative measurement, the difference between the apices of the four balloons and the flat substrate was measured.

### 2.5. Flow-Field Measurements of VIF with Particle Image Velocimetry

A micro-particle image velocimetry (PIV) technique was used to measure the flow field around the microballoons (the experimental setup is shown in Figure 3 and Appendix A). After fixing the device on an XY-piezo drive stage (ML-20XYL, MESS-TEK, Saitama, Japan), 8 µL of water containing fluorescent tracer beads with a diameter of 1 µm at a concentration of 9.1 × 10^8^/mL (17154-10, Polysciences, Inc., Warrington, PA, USA) was dropped on the microballoon surface. The substrate was then covered with a cover glass. Here, the thickness of the reagent space was set to approximately 200 µm by spacers in the bottom layer.

Two sinusoidal wave signals with a 90° phase offset were applied to the piezo stage using the waveform generator (AG 1022F, OWON, Zhangzhou, China) via an amplifier (M 2501-1, MESS-TEK, Japan) to induce circular vibrations in the horizontal plane. The vibration conditions were configured to exhibit a frequency of 600 Hz and an amplitude of 3.3 µm.

The flow field around the balloon actuated with different pressures was determined using PIV from continuous images captured using a CMOS camera (DFK33UX174, The Imaging Source, Inc., Bremen, Germany) at a frame rate of 100 fps with an exposure time of 1/50,000 s (20 μs) through a high NA objective lens (UCPlanFLN 20x/0.70, Olympus, Inc., Tokyo, Japan). Based on the acquired images, the velocity field was generated by correlating small regions of the particle images using PIV software (Flow Expert, KATO Koken, Kanagawa, Japan). A directional cross-correlation method was used to obtain velocity vectors. The size of the interrogation window was set to 48 pixels × 48 pixels (15.8 μm × 15.8 μm in the physical dimension, 0.33 μm/pixel).

For flow visualization, the movement of the tracer particles was visualized by superimposing the obtained continuous images in 0.1 or 0.2 s using image processing software (ImageJ, v1.8.0).

## 3. Results and Discussion

### 3.1. Characterization of the Balloon Actuator

First, we verified the actuation of the balloon device under air pressure and measured its displacement. Figure 4 shows the top layer of the device with a 13 µm thick elastomer membrane. In the absence of pressure, the elastomer membrane remained flat (Figure 4a). When a pressure of 10 kPa was applied to inlets A and B, dome-shaped microballoons appeared in the 5 × 5 array (Figure 4b). With an increase in pressure from 0 to 30 kPa, we observed an increase in balloon displacement (Figure 4c,d). The expanded thin membrane returned to its originally flat state when the pressure was removed, and protrusions occurred again when the pressure was reapplied (Appendix A). Figure 4e,f show the selective actuation of the microballoon columns when air pressure was applied from each inlet. A pneumatic pressure of 30 kPa was applied to inlets A and B to confirm whether the corresponding balloons were activated. When air pressure was applied to inlet A, the microballoons at the center and outermost columns in the 5 × 5 array were inflated. When air pressure was applied to inlet B, the two balloon columns between the previous columns of inlet A (Appendix A) were inflated. Figure 4g shows the relationship between the applied pressure and the displacement of the center of the balloon. Balloon displacements of ~20 µm at 10 kPa and ~38 µm at 30 kPa were achieved. The pressure-deformation curve within this range was nonlinear, following the typical Neo-Hookean-type response of rubber.

These results demonstrate the successful operation of the proposed microballoon actuation system. The displacement of the inflated microballoons was controlled by applying pressure. In addition, we confirmed that this microballoon could be inflated with application and returned to the original flat shape with the removal of pressure, within a range of approximately 0–30 kPa, with no visible plastic deformation. Each microballoon column connected to a different inlet was actuated by selectively applying pressure to each inlet, facilitating controlled actuation using the multichannel inlet system.

### 3.2. Induction of VIF around the Balloon Actuator

Next, the flow field generated around a microballoon under circular vibration was examined. Figure 5 shows an image of the movement of fluorescent tracer particles at the *z* = 35 µm horizontal plane above the bottom surface, in which the 20 consecutive images captured at a rate of 100 fps were superimposed. Without air pressure, the particles in the medium above the flat substrate exhibited negligible movement (Figure 5a). In contrast, a swirling flow was generated above the microballoon when a pressure of 30 kPa was applied (Figure 5b). This flow occurred from the outer edge of the microballoon towards its center (Appendix A).

Next, the flow field around the microballoon was quantitatively measured using PIV. The analysis results for the *z* = 35 µm plane are shown in Figure 6a–c as 2D vector fields with applied pressures of 0, 20, and 30 kPa, respectively. Without applying pressure (0 kPa), no flow was induced (Figure 6a). However, when a pressure of 20 kPa was applied, flow whirling occurred around the balloon (Figure 6b). As the pressure increased to 30 kPa, a flow velocity as high as ~400 µm/s, equivalent to the velocity induced around the solid cylindrical micropillars [15,37], was observed (Figure 6c). Figure 6d shows the velocity distribution in the radial direction (*r*) from the center of the balloon at applied pressures of 0, 10, 20, and 30 kPa. The magnitude of the maximum velocity increased as the applied pressure increased. Furthermore, the peak position at 20 kPa was located at approximately *r* = 60 µm, whereas it was at approximately *r* = 80 µm at 30 kPa. These results reflect the fact that the peak position shifts outward as the balloon expands in the vertical and lateral directions.

Flow field measurements were also conducted at various vertical cross-sections to reveal the three-dimensionality of the flow patterns surrounding the dome-shaped microballoon (Appendix A). Within the heights ranging from the base up to the vertex of the microballoon (*h* = 0–39 µm), a flow direction converging towards the center was observed, similar to the particle movement depicted in Figure 5b. In contrast, a flow radiating outward from the center was observed at positions above the microballoon (*h* > 52 µm). This pattern indicates a three-dimensional flow field encompassing the microballoon. Specifically, while swirling, the flow originating from the bottom substrate converged towards the vertex of the microballoon and subsequently spread outward from the vertex at a higher elevation, as depicted in Appendix A.

These results confirm that a flow field can be actively controlled by combining pneumatic control and vibration. The inflation of microballoons by the applied pressure significantly influenced the surrounding fluid, inducing a net flow under vibration. In addition, our flow field measurements at various heights indicate the presence of a pronounced three-dimensional flow pattern similar to that observed in our previous study using a cylindrical micropillar [37].

### 3.3. Spatial Control of Flow Patterns

Finally, the potential applications of microballoon actuation for spatial flow control were verified. To achieve this, we observed the flow field with selective actuation of the microballoons. Figure 7a shows a series of superimposed images of tracer beads above a pair of balloon actuators (the actual video is shown in Appendix A). Each balloon was connected to inlet A or B. Upon applying pressure to inlet A, a swirling flow pattern was observed only on the right side of the image (3 s). Immediately after the pressure application was terminated, the swirling flow on the right side of the image disappeared (6.6 s). Subsequently, when inlet B was activated, swirling flow occurred only on the left side (8 s). After stopping the pressure application at 12 s, pressure was simultaneously applied to both inlets A and B, resulting in flow induction around the microballoon on both the right and left sides (14 s). Spatial control of the flow field around the microballoons could be achieved by toggling the on-off state of the pressure application while the vibration remained active. The flow field under periodic switching was measured using PIV. Figure 7b,c show the 2D vector plots of the velocity field above a pair of balloon actuators. Similar to Figure 7a, a swirling flow was observed on either the right or left side, depending on the pressure applied from inlet A or B.

These results demonstrate that the flow field can be dynamically altered by selectively applying pressure to each inlet. The developed device and periodic operation have the potential to function as a micromixing of minute fluids with periodic switching of flow patterns, reminiscent of blinking vortex flows [33,34].

## 4. Conclusions

For the active control of the VIF, we developed a microballoon actuation system with an elastomer membrane driven by air pressure. Although this type of actuator was developed for other purposes [28,29,30], its application was mainly limited to millimeter-scale balloons, rendering them unsuitable for microscale fluid control using VIF. In this study, we employed a process that involves spin-coating elastomer membranes, followed by forming through-holes using the DRIE technique. This enables the fabrication of actuators with thin and small-diameter membranes that can generate displacements at relatively low pressures. Additionally, compared with previous studies involving the spatio-temporal control of fluid manipulation using mobile microbubbles [27], our system provides versatility using a relatively simple experimental setup.

Measurements of the pressure range and displacement achievable by the proposed balloons revealed that displacements of up to 38 µm could be achieved within the range of 0 to 30 kPa. This value is compatible with typical pumping systems used in microfluidic control, signifying the ease of implementation. Furthermore, we investigated VIF generation around the microballoon and measured the flow field. With microballoon inflation, the on-off state of the flow field can be controlled. This technique further enables control of the maximum flow velocity and peak position of the flow field based on the magnitude of the applied pressure. The induced velocity values are comparable to those observed in previous studies on VIF using solid structures, suggesting that the proposed system can be applied to flow control in microenvironments, such as particle manipulation and fluid mixing [15,35]. Additionally, flow field measurements at various vertical cross-sections revealed a strong three-dimensional circulating flow field along the microballoon surface.

Finally, we successfully achieved selective actuation of the microballoons using a multichannel air pressure system, facilitating on-demand flow pattern control. This capability offers selective actuation of microballoons based on the selection of the air pressure inlet. This system has potential applications, such as mixing through spatio-temporal control of the vortex position to reproduce the blinking vortex phenomenon [33,34] and the sophisticated manipulation of microobjects, particularly in their transport [16], by controlling the temporal variation of microballoon spacing. This capability is expected to provide promising applications in microfluidics with the active control of flows by optimizing the arrangement and size of microballoons.

## Figures and Tables

**Figure 1 micromachines-14-02010-f001:**
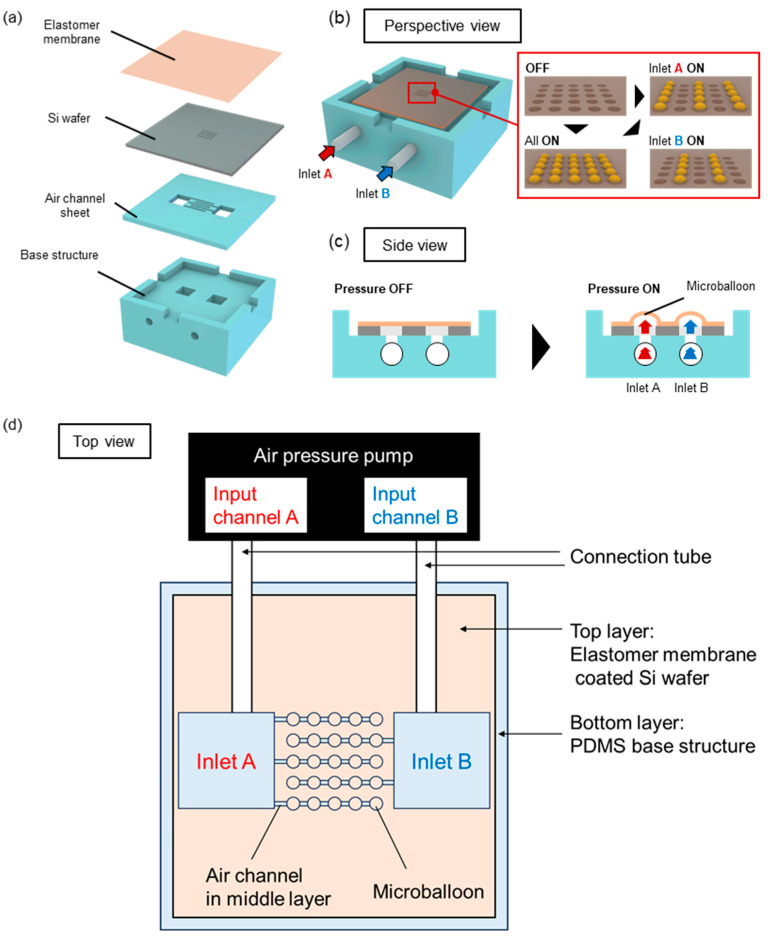
(**a**) Schematic of structures and assembly of the device. (**b**,**c**) Principle of pneumatic actuation of microballoons in (**b**) perspective and (**c**) side views. (**d**) Connection scheme of air inlets and air channels to drive selected columns of microballoons.

**Figure 2 micromachines-14-02010-f002:**
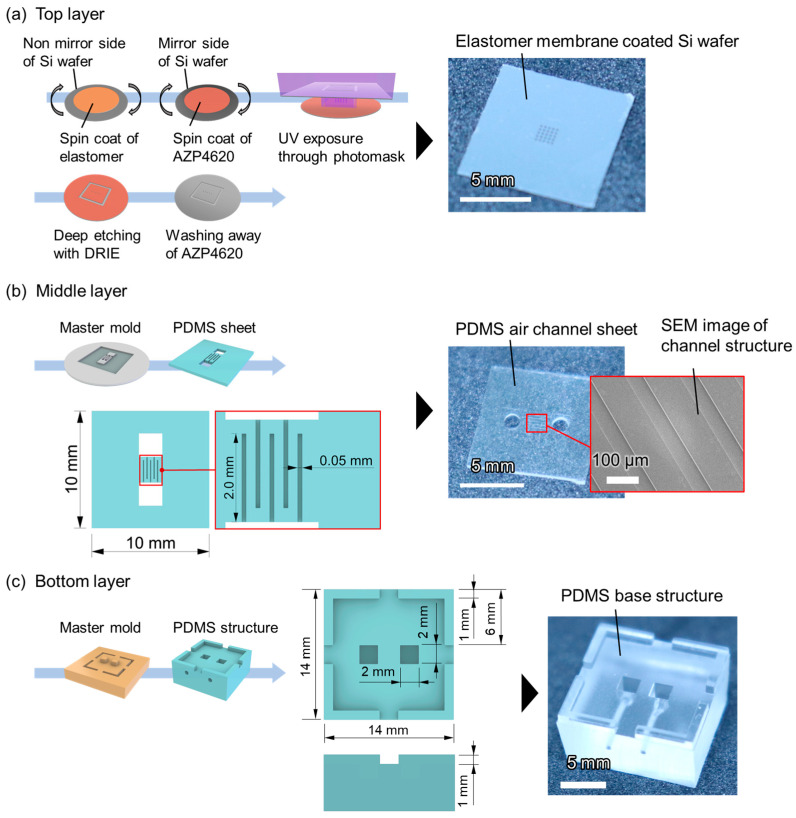
Fabrication process, dimensional drawing, and actual image of each layer consisting of balloon actuators. (**a**) Top layer: Si wafer with through-holes, over which the elastomer membrane was coated. (**b**) Middle layer: PDMS sheet with air channels to apply air pressure. (**c**) Bottom layer: PDMS base structure for tubing.

**Figure 3 micromachines-14-02010-f003:**
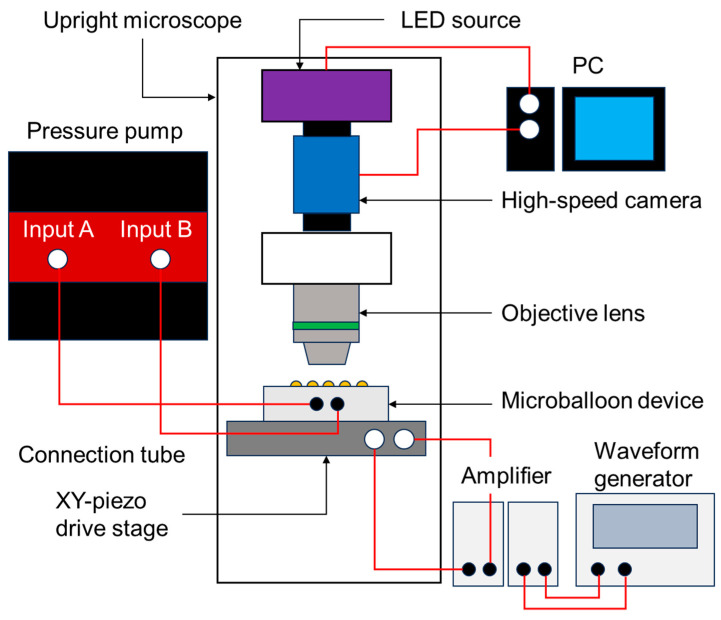
Schematic of the experimental setup for the VIF measurement with micro-PIV.

**Figure 4 micromachines-14-02010-f004:**
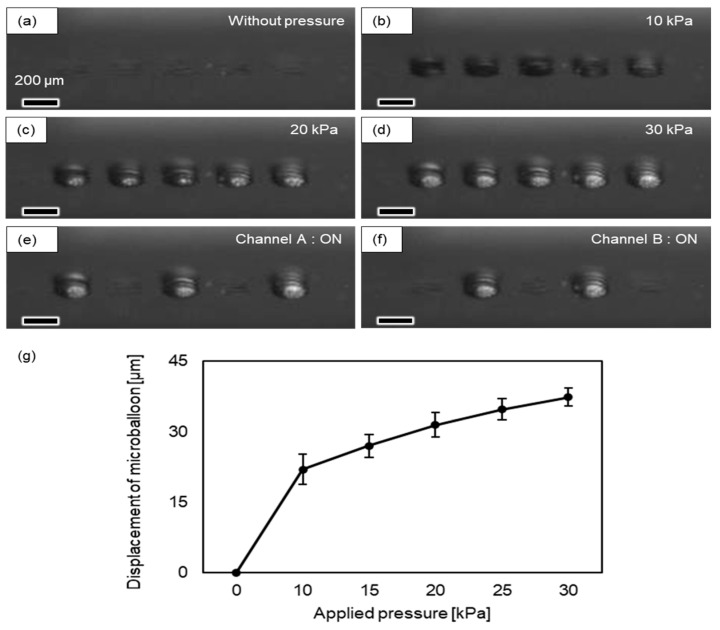
Actuation test of microballoons with applied air pressure. (**a**–**d**) Horizontal (but slightly slanted) observations of balloon actuators when (**a**) 0, (**b**) 10, (**c**) 20, and (**d**) 30 kPa pressures were applied to all balloons. (**e**,**f**) Selective actuation of columns of actuators when pressure is applied to (**e**) inlet A and (**f**) inlet B. (**g**) Relationship between applied pressure and balloon displacement measured using a laser scanning microscope. The average and standard deviation of triplicated experiments are plotted.

**Figure 5 micromachines-14-02010-f005:**
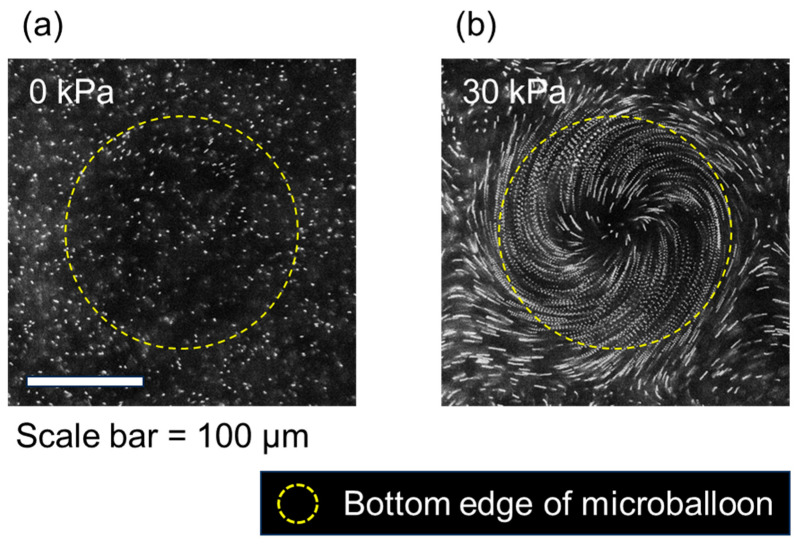
Fluorescent images of tracer particles around the microballoon superimposed for 0.2 s under applied pressures of (**a**) 0 kPa and (**b**) 30 kPa subjected to circular vibration.

**Figure 6 micromachines-14-02010-f006:**
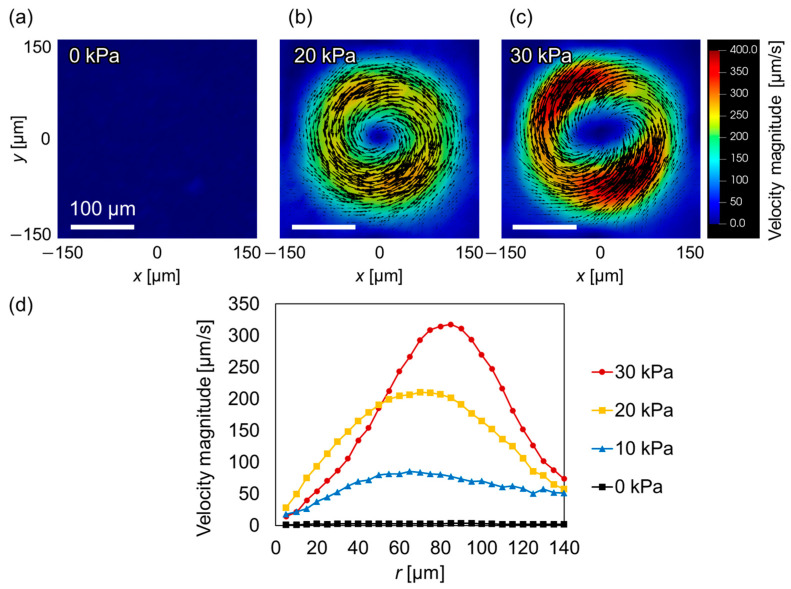
Two-dimensional vector plots of velocity fields measured using PIV in the *z* = 35 µm plane around the microballoon under pressures of (**a**) 0 kPa, (**b**) 20 kPa, and (**c**) 30 kPa subjected to circular vibration. (**d**) Azimuthal velocity profiles as a function of distance from pillar center (*r*) under various applied pressures.

**Figure 7 micromachines-14-02010-f007:**
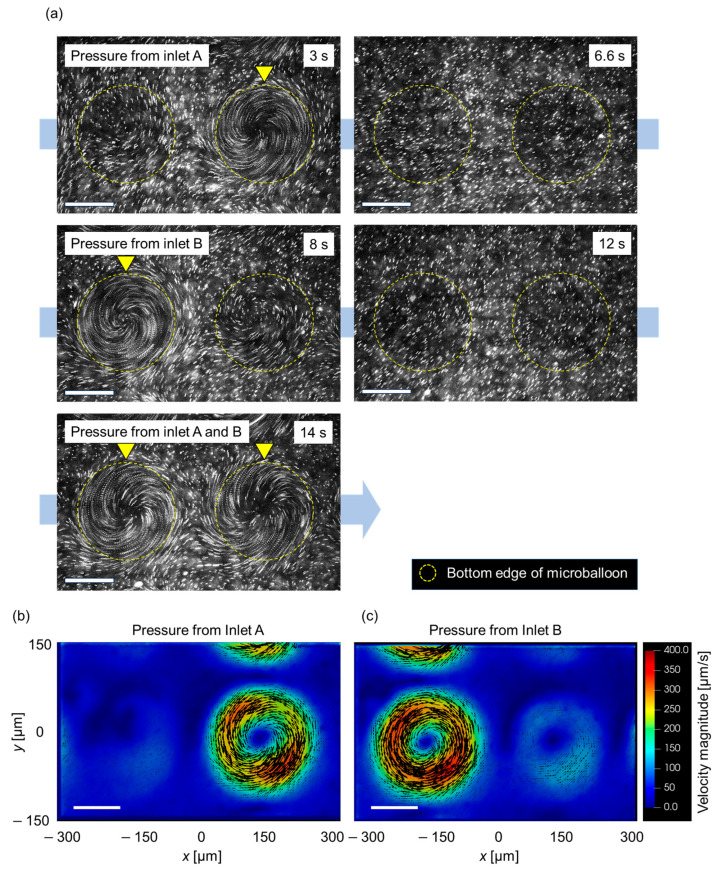
(**a**) Series of fluorescent images of tracer particles around the microballoon under an applied pressure of 30 kPa with circular vibration. Each raw image is superimposed with the original captured image for 0.1 s. (**b**,**c**) Two-dimensional vector plots of velocity field measured with PIV in *z* = 35 µm plane around microballoon under air pressure from (**b**) inlet A and (**c**) inlet B. The scale bar in the figures represents 100 µm.

## Data Availability

The datasets generated and analyzed in this study are available from the corresponding author upon reasonable request.

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
