# Peer review of "Pneumatic Microballoons for Active Control of the Vibration-Induced Flow"

_micromachines, 2023, doi:10.3390/mi14112010_

Round 1

Reviewer 1 Report

Comments and Suggestions for Authors

In this paper, the authors try to improve the device of the vibration-induced flow (VIF) using elastic membrane protrusions (microballoons) actuated by pneumatic pressure. Instead of having a complicated design, the device allows to change the follow patterns by changing the places of actuated microballoons within one device, which was difficult for previous VIF devices. This paper would be the worthy of publication at Micromachines. However, I encourage resubmission following extensive experiments/revisions.

1)     Can the authors explain why the size and spacing of circles are 200 μm diameter and 300 μm, respect. Also, can the authors explain that the array with square of the microballoons is optimal design. Instead of the square, do you consider the design with array hexagonally. These parameters are important to control flow patterns.

2)     In Figure 3, the authors should add how many times did their experiments since the graph has error bar.

3)     In Figure 4, can the authors explain whether you can contrail the flow of tracer particles with  clockwise or counterclockwise.

4)   The authors need to explain more about applications. They need to explain how can this device be used to transport, concentrate, and rotate for example cells and particles like previous VIF devices?

Author Response

Author's Reply to the Review Report (Reviewer 1)

In this paper, the authors try to improve the device of the vibration-induced flow (VIF) using elastic membrane protrusions(microballoons) actuated by pneumatic pressure. Instead of having a complicated design, the device allows to change the follow patterns by changing the places of actuated microballoons within one device, which was difficult for previous VIF devices. This paper would be the worthy of publication at Micromachines. However, I encourage resubmission following extensive experiments/revisions.

Response: Thank you for reviewing and providing us insightful comments to strengthen our paper.

1)

Can the authors explain why the size and spacing of circles are 200 μm diameter and 300 μm, respect. Also, can the authors explain that the array with square of the microballoons is optimal design. Instead of the square, do you consider the design with array hexagonally. These parameters are important to control flow patterns.

Response: Thank you for your comment. For the diameter of the microballoon, we selected 200 µm, which is larger than the typical 100 µm diameter used for solid pillars in our previous study. We made this choice because increasing the hole diameter may result in greater balloon deformation. This increased deformation leads to a wider range of flow effects, potentially enhancing the effectiveness of active control of microballoons. Additionally, precise alignment of through-holes in the top layer and air channels in the middle layer is crucial for the selective control of microballoons. When the through-hole size is too small, achieving accurate alignment becomes challenging, leading to reduced reproducibility in device fabrication. On the other hand, if the microballoon diameter is too large, it becomes difficult to observe under a microscope, and we cannot fully leverage the advantages of our micro-fabrication techniques. Regarding the spacing of microballoons at 300 µm, in this design, the distance of the outer bottom edge between adjacent balloon is 100 µm. in this spacing, interference of flow occurs between adjacent microballoons. For this study, we wanted to select conditions that make the flow interference between adjacent balloons more likely to assess the effect of selective actuation. Hence, we chose this balloon spacing. As for the arrangement of the pillars, yes, we did consider adopting a hexagonal arrangement. However, in this study, our primary focus is on demonstrating the microballoon device. Therefore, we opted for a square array, which makes selective actuation through a multichannel inlet system easier. Nevertheless, as you rightly pointed out, adopting a hexagonal array is one method to alter flow patterns. We believe that complex arrangements such as a hexagonal array can be realized by optimizing the selective actuation system and the fabrication process. We have provided further comments and explanations regarding this design selection in Section 2.2..

2)

In Figure 3, the authors should add how many times did their experiments since the graph has error bar.

Response: Thank you for pointing out. The experiment was conducted three times. We have included in the caption of Figure 3 that the average and standard deviation of triplicated experiments are plotted.

3)

In Figure 4, can the authors explain whether you can contrail the flow of tracer particles with clockwise or counterclockwise.

Response: Yes, we think it is possible to control the flow with either a clockwise or counterclockwise swirling flow. The direction of rotation in swirling flow depends on the direction of circular vibration (reference numbers 8 and 15). In our present study, we induced counterclockwise swirling flow around the microballoon by applying counterclockwise circular vibration to the device. However, we can generate a circular flow in the opposite direction by applying clockwise circular vibration.

4) The authors need to explain more about applications. They need to explain how can this device be used totransport, concentrate, and rotate for example cells and particles like previous VIF devices?

Response: Thank you for your insightful comment. One potential application is micromixing with periodic switching of flow patterns, as mentioned in the main text. Furthermore, we believe that this device's advantage of selective actuation can be leveraged for various applications, especially in manipulating microobjects, particularly in their transport. Whether microobjects are transported between pillar arrays or circulate a single pillar depends on the pillar spacing and the size of the microobjects (reference number 6). By utilizing the selective actuation of this device to control the temporal variation of microballoon spacing, we can anticipate the ability to regulate the transport state of microobjects. These descriptions are included in the Conclusion.

Reviewer 2 Report

Comments and Suggestions for Authors

This study presents a microdevice that actively controls vibration-induced flow (VIF) patterns by using elastic microballoons actuated through pneumatic pressure. It offers precise spatial and temporal fluid manipulation, achieving displacements up to 38 μm in the 0-30 kPa pressure range. The device can actively manipulate flow fields, induce swirling flows, and selectively actuate microballoons via a multi-input channel system. This technology has promising applications in advanced microfluidics, including fluid mixing and particle manipulation. This review suggests accepting the manuscript after minor revision.

Questions

1. In Fig 1 (a), the membrane is not clearly visible on top of the Si wafer. Maybe a different color of the membrane should be used.

2. Fig 1 (b) should show the position of the piezo stage; then, it will be much easier for the reader to understand the vibration source from the scheme.

3. Line 225 of Page 7, why the microballon is "reversibly actuated"? Do you mean "repeatedly"?

4. Line 266 of Page 8, can you roughly explain the mechanism of the swirling flow? How does it form?

5. Line 271 of Page 8, ref[35] was not from the same group. Maybe you mean ref[37]?

6. Line 297 of Page 9, is it possible to show by experiment that the swirling flow can be used for micromixing?

Author Response

Author's Reply to the Review Report (Reviewer 2)

This study presents a microdevice that actively controls vibration-induced flow (VIF) patterns by using elastic microballoons actuated through pneumatic pressure. It offers precise spatial and temporal fluid manipulation, achieving displacements up to38 μm in the 0-30 kPa pressure range. The device can actively manipulate flow fields, induce swirling flows, and selectively actuate microballoons via a multi-input channel system. This technology has promising applications in advanced microfluidics, including fluid mixing and particle manipulation. This review suggests accepting the manuscript after minor revision.

Response: Thank you for reviewing and providing us insightful comments to strengthen our paper.

Questions

  1. In Fig 1 (a), the membrane is not clearly visible on top of the Si wafer. Maybe a different color of the membrane should be used.

Response: Thank you for pointing out. To clearly distinguish between the elastomer membrane and the Si wafer, we have utilized different colors for the membrane and the Si wafer. Additionally, both layers have been separated to highlight them as distinct layers. You can find this modification in Figure 1a of the revised manuscript.

  1. Fig 1 (b) should show the position of the piezo stage; then, it will be much easier for the reader to understand the vibration source from the scheme.

Response: Thank you for your comment to improve our manuscript. Figure 1 primarily focuses on depicting the driving principle for selective actuation within the microballoon device, and therefore, it does not include an explanation of the vibration system. However, as you pointed out, explaining of the vibration source is crucial for the reader’s comprehension. To address this, we have made an enhancement by incorporating Figure 3 in Section 2.5, which represents a schematic of the experimental setup for VIF measurement with micro-PIV. This addition aims to provide a more comprehensive understanding of the VIF system. In addition, to further enhance clarity, we have incorporated an actual image of the microballoon device affixed to an XY-piezo drive stage in Supplementary Figure S5.

  1. Line 225 of Page 7, why the microballon is "reversibly actuated"? Do you mean "repeatedly"?

Response: Thank you for your comment. In this context, we intended to describe the characteristics of the microballoon, which can be actuated repeatedly without any visible plastic deformation. We have adjusted the terminology employed within Section 3.1 of the manuscript to align with this clarification.

  1. Line 266 of Page 8, can you roughly explain the mechanism of the swirling flow? How does it form?

Response: Thank you for your comment. As we discussed in the Introduction, a localized flow, such as the swirling flow, is initiated through relative periodic oscillation involving solid-liquid interaction. When an infinite planar substrate oscillates parallel to its surface, only a transient velocity field with a zero mean is generated.

  1. Line 271 of Page 8, ref [35] was not from the same group. Maybe you mean ref[37]?

Response: Thank you for pointing out. We have corrected the reference number in Line 271 of Page 8 as [37]. We have meticulously checked the references list again.

  1. Line 297 of Page 9, is it possible to show by experiment that the swirling flow can be used for micromixing?

Response: Thank you for your insightful comment. Indeed, we concur with your perspective. However, demonstrating and experimental evidence for two co-flowing fluids mixing, commonly employed in flow-based mixers, poses challenges in this system due to the difficulty of maintaining a consistent initial state. As a result, the primary focus of this research endeavor is centered on detailing the fabrication and demonstration of the microballoon actuator. Currently, mixing using fixed pillars is being examined in our separate ongoing work. However, the pursuit of achieving enhanced mixing performance through active flow control remains a subject for further investigation.